# ♻ Recycling the Web: A Method to Enhance Pre-training Data Quality and Quantity for Language Models

**Thao Nguyen**[1,2*]  **Yang Li**[1]  **Olga Golovneva**[1]
**Luke Zettlemoyer**[1,2]  **Sewoong Oh**[2]  **Ludwig Schmidt**[3]  **Xian Li**[1]
[1]FAIR at Meta   [2]University of Washington   [3]Stanford University

## Abstract

Scaling laws predict that the performance of large language models improves with increasing model size and data size. In practice, pre-training has been relying on massive web crawls, using almost all data sources publicly available on the internet so far. However, this pool of natural data does not grow at the same rate as the compute supply. Furthermore, the availability of high-quality texts is even more limited: data filtering pipelines often remove up to 99% of the initial web scrapes to achieve state-of-the-art. To address the "data wall" of pre-training scaling, our work explores ways to transform and recycle data discarded in existing filtering processes. We propose REWIRE, **RE**cycling the **W**eb with gu**I**ded **RE**write, a method to enrich low-quality documents so that they could become useful for training. This in turn allows us to increase the representation of synthetic data in the final pre-training set. Experiments at 1B, 3B and 7B scales of the DCLM benchmark show that mixing high-quality raw texts and our rewritten texts lead to 1.0, 1.3 and 2.5 percentage points improvement respectively across 22 diverse tasks, compared to training on only filtered web data. Training on the raw-synthetic data mix is also more effective than having access to $2\times$ web data. Through further analysis, we demonstrate that about 82% of the mixed in texts come from transforming lower-quality documents that would otherwise be discarded. REWIRE also outperforms related approaches of generating synthetic data, including Wikipedia-style paraphrasing, question-answer synthesizing and knowledge extraction. These results suggest that recycling web texts holds the potential for being a simple and effective approach for scaling pre-training data. We make our high-quality synthetic data publicly available at https://huggingface.co/datasets/facebook/recycling_the_web.

## 1 Introduction

Over the past few years, large language models (LLMs) have rapidly improved on various benchmarks. This progress was driven largely by scaling up model size, training FLOPs, and in particular, dataset size (Hoffmann et al., 2022). For instance, Llama-3 was pre-trained on 15T tokens sourced from publicly available data (Grattafiori et al., 2024), while the previous generation of Llama models was only trained on 2T tokens (Touvron et al., 2023). The vast quantity of training tokens so far is obtained primarily from internet crawls containing billions of web pages (Penedo et al., 2023; Weber et al., 2024; Soldaini et al., 2024), made publicly available by Common Crawl.

While compute resources can scale in accordance with scaling laws and improved hardware efficiency, the growth of public human-generated texts has been less sustainable (Longpre et al., 2024). Villalobos et al. (2024) posit that the current rate of LLM development will exhaust the available stock of internet data between 2026 and 2032. Despite growing concern that LLM pre-training is hitting such a "data wall", existing work on data curation still finds it necessary to discard the majority–sometimes up to 99%–of the data collected to ensure quality and state-of-the-art downstream performance (Li et al., 2024; Penedo et al., 2024). As

---

*Correspondence to thaottn@cs.washington.edu.

we approach the "data wall" while throwing away 99% of web-crawled data, a fundamental question arises: *can we recycle documents that have been discarded by quality filters to make them useful for pre-training?*

Existing work has started exploring different directions to address the impending data bottleneck. For example, we can go beyond the public internet data and obtain licensed, hard-to-access sources (e.g., Reddit and news sites). However, while on average web crawls are of lower quality than these curated sources, previous research has shown that after enforcing quality control, the former can still dominate the latter in terms of token quantity and contribution to downstream performance (Xie et al., 2023). Another line of work proposes relaxing or changing the curation strategies to recover raw documents that have been removed by previous quality filters (Su et al., 2024; Muennighoff et al., 2023). In addition, generating synthetic data for certain skills or formats has also been studied to increase the token availability (Gunasekar et al., 2023; Li et al., 2023; Maini et al., 2024).

Our work combines the two aforementioned strategies: synthetic data generation and recycling discarded documents. We propose **RE**cycling the **W**eb with gu**I**ded **RE**write (**REWIRE**), which involves taking all documents that are of moderate quality (i.e., having passed some rule-based filters), using an LLM to identify the purpose of the text content, and then asking the LLM to come up with an improved

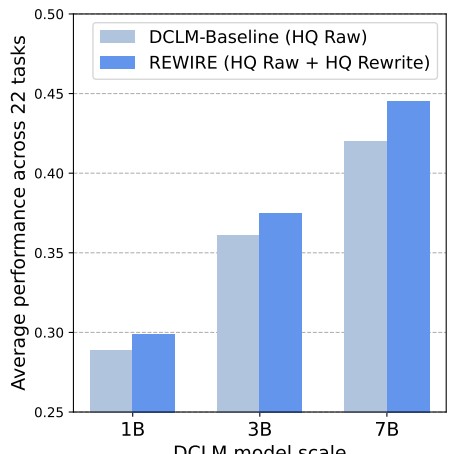

Figure 1: **REWIRE offers increasing performance gains as we scale up model size and training token budget.** Our experiments simulate the setting in which high-quality texts are limited and the large token budget (set to be Chinchilla-optimal in this figure) necessitates training on the same filtered dataset multiple times (i.e., 4 epochs). On average across 22 tasks from DCLM's CORE (Li et al., 2024), mixing in the same amount of synthetic data as that of high-quality web data ("HQ Raw + HQ Rewrite") consistently outperforms training on only the latter ("HQ Raw").

document conditioned on chain-of-thought reasoning. Unlike most existing work on synthetic data, our approach specifically targets the vast quantity of low-quality documents that are somewhat informative but still not considered high-quality by existing filters. We use LLM's knowledge and reasoning capabilities to recycle these documents and add them back to the training pool. The overall data generation pipeline is described in Figure 2.

Through extensive experiments at both 1B, 3B and 7B model parameter scales, we show that pre-training models on a combination of high-quality web-crawled data and high-quality rewritten data outperforms using the former alone (Section 3). Averaged across 22 tasks of the DataComp-LM benchmark (Li et al., 2024), the raw-synthetic data mixture improves performance by 1.0, 1.3 and 2.5 percentage points, at 1B, 3B and 7B scales respectively. The performance benefits of adding rewritten texts also hold across different ways of selecting high-quality raw documents. Furthermore, we demonstrate that the accuracy level achieved by combining raw and synthetic data matches that of using 2× more raw data (Table 1). We verify that our best baseline indeed contains a significant amount of synthetic data "recycled" from low-quality documents (Section 4.1).

Finally, we compare our rewritten data to three other variations of synthetic data from recent work (Maini et al., 2024; Su et al., 2024): extracted knowledge and diverse question-answer pairs synthesized from high-quality documents, Wikipedia-style rephrasing from low-quality ones. We show that **REWIRE** generates more diverse synthetic data (Section 4.3), which in turn contributes to higher performance on the DCLM benchmark (Table 1).

## 2 Experiment Setup

### 2.1 Data Pool

We seek to simulate *long token horizon training* (Su et al., 2024), a setting in which high-quality data is limited and the large token compute budget necessitates seeing the same samples

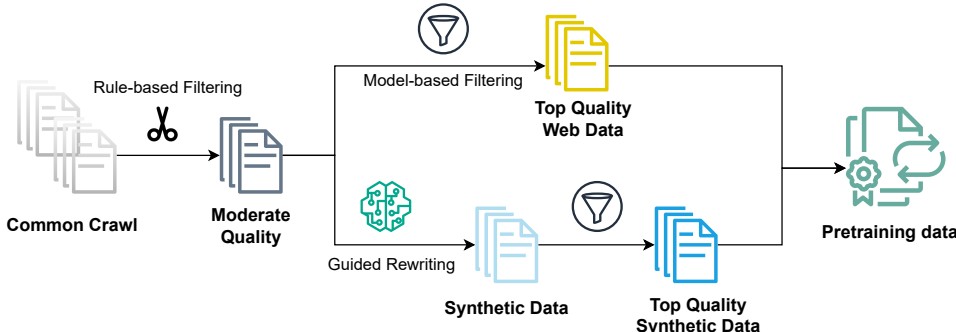

Figure 2: **The REWIRE pipeline.** We start with web documents from Common Crawl that has undergone some filtering (i.e., RefinedWeb heuristics (Penedo et al., 2023)), and thus are at least of moderate quality. State-of-the-art data curation approach, e.g. DCLM-Baseline (Li et al., 2024), applies further model-based filtering to retain only top-quality documents for pre-training. Our pipeline takes moderate-quality documents and prompts an LLM to do guided rewriting to generate improved versions of these documents. Finally, we select only high-quality synthetic documents and combine them with the DCLM-Baseline texts to form the final pre-training dataset.

multiple times during training. As Muennighoff et al. (2023) find that there are diminishing returns after four epochs compared to training on more unique tokens, we limit the number of sample repeats to be at most 4 in our main experiments. Appendix D contains additional experiments with larger training token budget, thus also making the data repetition rate higher (at most 8 - 10 times).

We fix the starting pool to be DCLM-RefinedWeb (Li et al., 2024), Common Crawl data that has passed the initial rule-based quality filters from RefinedWeb (Penedo et al., 2023) (e.g., repetition filter, page length filter, URL filter, etc.) and global deduplication, but has not gone through model-based filtering. With this pool of moderate-quality data, DataComp-LM (Li et al., 2024) futher selects only the top 10% based on scores from a fastText classifier (Joulin et al., 2016). This results in DCLM-Baseline.

Following the token budget set by DCLM at model sizes of 1B, 3B and 7B parameters, we fix the starting pool size to be 72B, 140B and 345B tokens respectively. This is to ensure that even after aggressive pruning, the high-quality data is repeated only at most 4 times. For instance, if the training budget is 28.8B tokens seen at 1B scale, choosing the top 10% based on fastText scores from a starting pool of 72B tokens would leave us with 7.2B tokens. We also experiment with relaxing the filtering threshold (e.g., by selecting the top 20% instead of top 10%) so that there are more unique tokens left after filtering.

## 2.2 Guided Rewriting

The central hypothesis of our **REWIRE** framework is that web documents contain diverse content and knowledge, but the writing structure can make them not coherent or elaborate enough to serve as informative pre-training examples. Inspired by recent work on leveraging the meta-cognitive capabilities of state-of-the-art LLMs (Didolkar et al., 2024), we prompt Llama-3.3-70B-Instruct (Grattafiori et al., 2024) to perform chain-of-thought reasoning on the original web document, such as identifying the task or purpose of the text, reasoning about the steps needed to achieve the purpose, etc. before generating an improved version of the original document. The full prompt we use can be found in Section B. We apply the same rewriting process to all documents in the starting pool (DCLM-RefinedWeb).

To control the quality of the generations, we further apply model-based filtering to the synthetic data (Figure 2). Following DCLM (Li et al., 2024), we train a fastText classifier (Joulin et al., 2016) on 400K documents split evenly between positive and negative classes. The positive data is the same as used in DCLM, which includes synthetic instruction data from OpenHermes 2.5 (Teknium, 2023) (OH-2.5) and high-scoring posts from the r/ExplainLikeImFive (ELI5) subreddit. The negative data are random samples selected from our rewriting generations. Similar to what was done to obtain DCLM-Baseline, we also aggressively filter all the rewritten outputs and only use the top 10% based on the scores of our customized fastText classifier.

## 2.3 Training & Evaluation

Following the same protocol as DCLM, we fix the training hyperparameters and total budget (i.e., number of samples seen) to match what was reported in previous work (Li et al., 2024). We train all models using the Lingua framework (Videau et al., 2024). We mainly experiment with 1B-1x, 3B-1x and 7B-1x model scales (1x refers to the Chinchilla multiplier), using Llama-2 architecture and tokenizer as the backbone (Touvron et al., 2023). We set the model parameters (e.g., number of layers, number of heads) to be the same as DCLM's. More training details can be found in Appendix A.

For evaluation, we report the same metrics as DCLM, i.e. MMLU *5-shot accuracy* (Hendrycks et al., 2020) as well as CORE *centered accuracy* averaged over 22 tasks (e.g., HellaSwag (Zellers et al., 2019) and ARC-easy, ARC-challenge (Clark et al., 2018)). To compute centered accuracy, each task's performance is linearly rescaled so that 0 corresponds to random guessing and 1 corresponds to perfect accuracy. Li et al. (2024) have shown that CORE metric offers a low-variance signal even at small scales. More descriptions of the 22 tasks can be found in the DCLM paper.

## 3 Results

We provide our main result in Table 1, which demonstrates that **REWIRE** achieves the best average performance across 22 tasks of CORE.

### 3.1 Baselines

As described in Section 2.1, we start with a fixed pool of data randomly sampled from DCLM-RefinedWeb (Li et al., 2024) and filter with DCLM's fastText classifier to select the highest-quality documents. For comparison with another variation of high-quality web texts, we also experiment with the data released by PreSelect (Shum et al., 2025), which is curated from the same pool, DCLM-RefinedWeb, but with a different fastText classifier trained to classify a document's predictive strengths of model downstream capabilities. For comparison with other variations of synthetic data, we experiment with the data released by Nemotron-CC (Su et al., 2024)[1], which contains multiple augmented versions of the same web-crawled data pool, generated using different prompts depending on how high-quality the original web document is.

Below we describe in details the baselines from Table 1:

- `Raw text (top 10%)`: We rank examples from the starting pool by the scores from DCLM's fastText classifier (Li et al., 2024) and select the top 10% highest-scoring texts. This results in the same data distribution as the final `DCLM-Baseline` pool published by DCLM.

- `Raw text (top 20%)`: Here the fastText filtering threshold is relaxed, allowing for relatively more unique tokens to be included which could benefit multi-epoch training. This is reflected in the final dataset size being double that of `Raw text (top 10%)`.

- `Rewritten text (top 10%)`: We rank all the synthetic data resulting from guided rewriting by the scores from our fastText classifier trained in a similar fashion to DCLM's (Section 2.2). We then select top 10% of the rewritten texts.

- `Raw text (top 10%) + Rewritten text (top 10%)`: We combine the highest-quality texts from the original starting pool as well as the same pool after being transformed with guided rewriting. This means that some of the selected documents will have both the web-scraped and the rewritten versions included, while some will only have either version. We analyze the overlap between the two distributions later in Section 4.1.

- `PreSelect/ PreSelect + Rewritten text (top 10%)`: Since the data released by Shum et al. (2025) is already filtered to be the top 10% of the DCLM-RefinedWeb pool based on their quality metric, we experiment with training on the open-source curated data directly, as well as mixing it with our highest-quality rewritten data.

- `Nemotron-CC High-quality (HQ) diverse QAs`: Su et al. (2024) prompt an LLM to ask questions in various forms (e.g., yes/no question, open-ended question, multi-choice

---

[1]https://data.commoncrawl.org/contrib/Nemotron/Nemotron-CC/index.html

question) about factual information in a document and provide the correct answers. They apply this prompt only to high-quality web texts from DCLM. Since the open-source data for Diverse QAs split already contains the raw documents followed by QAs, we randomly sample data from the split until the token count from the raw texts (excluding QAs) matches the token budget. We note that despite starting from the same pool (DCLM), due to differences in filtering criteria, Nemotron-CC HQ web documents could differ significantly from the documents selected for the `Raw text (top 10%)` baseline.

- `Raw text (top 10%) + Nemotron-CC HQ extracted knowledge`: For this synthetic data variation, Su et al. (2024) prompt LLMs to convert existing knowledge in the raw text to some standard technical formats (i.e., textbooks and Wikipedia) and discard uninformative content (i.e.,"only restate what is already in the text"). The open-source data for this split does not contain the corresponding original documents, so we combine `Raw text (top 10%)` with the extracted knowledge synthetic data. As previous work only applies this prompt to high-quality documents, which are assumed to be limited in quantity, the number of tokens generated from extracted knowledge therefore is also limited. To simulate this setting, we fix the number of extracted knowledge samples to be the same as the number of documents in `Raw text (top 10%)`.

- `Raw text (top 10%) + Nemotron-CC Medium-quality (MQ) Wikipedia rephrasing`: For relatively lower quality documents from DCLM, Su et al. (2024) follow the method proposed by Maini et al. (2024) and use the LLM to solely change the writing style to be Wikipedia-like, instead of using LLM "as a knowledge bank". This is similar to our approach in the sense that the synthetic data comes from a disjoint set of documents that are not selected for pre-training. We randomly sample Wikipedia-rephrased segments until we reach the same token quantity as our `Rewritten text (top 10%)` baseline.

Our setup assumes a raw data bottleneck, i.e., the size of the starting pool is fixed. Given a limited number of moderate-quality (potentially usable) documents from this pool, we compare ways to filter and enrich the existing data. However, we also experiment with the setting where the starting pool size is doubled (e.g., moving from 72B to 144B tokens in total at the 1B scale, see the shaded rows in Table 1). If the stock of web data grew by twice as much (144B), what performance could we expect from aggressively filtering raw data alone, and could synthetic data help close the performance gap at the current data scale (72B)?

### 3.2 Performance on DCLM benchmark

In Table 1, we find that while training on synthetic data alone still lags behind training on highest-quality raw texts, using *mixed* data distributions (i.e., `Raw text (top 10%) + Rewritten text (top 10%)` or `PreSelect + Rewritten text (top 10%)`) outperforms using only the corresponding filtered web data. It is worth noting that our synthetic data significantly boosts MMLU performance without being rewritten specifically for this task (i.e., by converting into QAs or by selecting topics). At the 3B scale, mixing raw and synthetic data still improves MMLU, but tuning the mixing ratio becomes more important for improving average performance across a range of tasks. We note that the gain in average performance (relative to using only high-quality raw data from the same pool) increases with model and training scale: +1.0 percentage points (pp) at 1B-1x, +1.3pp at 3B-1x and +2.5pp at 7B-1x.

We also experiment with settings that simulate the large data regime beyond Chinchilla-optimal as are often adopted in practice (Touvron et al., 2023; Grattafiori et al., 2024). There, we train on the smallest-sized dataset for more than 4 epochs. Table 4 in Appendix D reports results for the *1B-5x: 144B tokens seen*, ~3-10 epochs setting, as well as the *7B-2x: 276B tokens seen*, ~4-8 epochs setting. The same findings hold: mixing rewritten data with high-quality web data brings significant improvement on both MMLU and CORE, e.g. +7.3% on MMLU and +2.3pp on average (at 7B scale) compared to training on DCLM-Baseline data alone.

Furthermore, we compare our rewritten data to three versions of Nemotron-CC (Su et al., 2024), which are representative, related approaches of synthetic data generation. We find that REWIRE consistently yields the best performance on average when mixing with highest-quality raw texts. Out of the three variations from Su et al. (2024), the extracted knowledge

| Baseline name | Pool size | Dataset size | MMLU↑ | CORE↑ |
|---|---|---|---|---|
| *1B-1x Setting: 28.8B tokens seen* | | | | |
| Raw text (top 10%), DCLM-Baseline (Li et al., 2024) | 72B | 7.2B | 0.266 | 0.289 |
| Raw text (top 20%) | 72B | 14.4B | 0.249 | 0.282 |
| Rewritten text (top 10%) | 72B | 7.2B | 0.266 | 0.270 |
| Raw text (top 10%) + Rewritten text (top 10%) | 72B | 7.2B + 7.2B | 0.268 | **0.299** |
| Raw text (top 10%), 2× | 144B | 14.4B | 0.252 | 0.294 |
| Raw text (top 20%), 2× | 144B | 28.8B | 0.243 | 0.291 |
| PreSelect (Shum et al., 2025) | 72B | 7.2B | 0.250 | 0.277 |
| PreSelect + Rewritten text (top 10%) | 72B | 7.2B + 7.2B | 0.239 | 0.284 |
| PreSelect, 2× | 144B | 14.4B | 0.258 | 0.284 |
| Nemotron-CC HQ diverse QAs (Su et al., 2024) | 72B | 7.2B | **0.299** | 0.284 |
| Raw text (top 10%) + Nemotron-CC HQ extracted knowledge | 72B | 7.2B + 2.7B | 0.250 | 0.295 |
| Raw text (top 10%) + Nemotron-CC MQ Wikipedia rephrasing | 72B | 7.2B + 7.2B | 0.248 | 0.285 |
| *3B-1x Setting: 55.9B tokens seen* | | | | |
| Raw text (top 10%), DCLM-Baseline (Li et al., 2024) | 140B | 14B | 0.251 | 0.362 |
| Raw text (top 20%) | 140B | 28B | 0.240 | 0.363 |
| Rewritten text (top 10%) | 140B | 14B | 0.286 | 0.317 |
| Raw text (top 10%) + Rewritten text (top 10%) | 140B | 14B + 14B | 0.285 | 0.364 |
| Raw text (top 10%) x 0.6 + Rewritten text (top 10%) x 0.4 | 140B | 14B + 14B | 0.274 | **0.375** |
| Raw text (top 10%), 2× | 280B | 28B | 0.256 | 0.369 |
| Raw text (top 20%), 2× | 280B | 55.9B | 0.254 | 0.360 |
| PreSelect (Shum et al., 2025) | 140B | 14B | 0.255 | 0.353 |
| PreSelect + Rewritten text (top 10%) | 140B | 14B + 14B | 0.310 | 0.367 |
| PreSelect, 2× | 280B | 28B | 0.253 | 0.360 |
| Nemotron-CC HQ diverse QAs (Su et al., 2024) | 140B | 14B | **0.380** | 0.363 |
| Raw text (top 10%) + Nemotron-CC HQ extracted knowledge | 140B | 14B + 5.3B | 0.247 | 0.364 |
| Raw text (top 10%) x 0.6 + Nemotron-CC HQ extracted knowledge x 0.4 | 140B | 14B + 5.3B | 0.261 | 0.364 |
| Raw text (top 10%) + Nemotron-CC MQ Wikipedia rephrasing | 140B | 14B + 14B | 0.258 | 0.360 |
| Raw text (top 10%) x 0.6 + Nemotron-CC MQ Wikipedia rephrasing x 0.4 | 140B | 14B + 14B | 0.268 | 0.368 |
| *7B-1x Setting: 138B tokens seen* | | | | |
| Raw text (top 10%), DCLM-Baseline (Li et al., 2024) | 345B | 34.5B | 0.326 | 0.420 |
| Raw text (top 10%) + Rewritten text (top 10%) | 345B | 34.5B + 34.5B | **0.447** | **0.445** |
| Raw text (top 10%), 2× | 690B | 69B | 0.356 | 0.425 |

Table 1: **Main results on the DCLM benchmark.** We report the performance of training with different datasets on MMLU and on average across 22 tasks of CORE (Li et al., 2024). Accuracies that are near random-chance performance are in gray. Across all three model and training budget scales, we observe that training only on high-quality synthetic data underperforms training on high-quality raw texts. However, combining these two subsets consistently boosts MMLU and overall performance, matching the accuracies of training on 2× more high-quality raw data (shaded rows). REWIRE is also more effective than other synthetic data variants (Su et al., 2024) at improving average performance.

format is the most helpful for increasing CORE performance. Even though Nemotron-CC's extracted knowledge and Wikipedia rephrasing do not help with MMLU, their diverse QAs are especially effective. This is potentially due to the alignment of the data format, as MMLU is made up of multiple-choice questions (Hendrycks et al., 2020). Overall these results suggest that REWIRE is more effective at complementing curated natural texts.

Finally, at all model parameter scales that we experiment with, combining carefully filtered raw and rewritten texts can match, if not outperform, the performance level of training on 2× more high-quality web documents, i.e. as if we had access to a starting pool with 2× more raw data. For instance, at the 1B model scale, using the `Raw text (top 10%) + Rewritten text (top 10%)` baseline from a starting pool of 72B tokens yields 26.8% accuracy on MMLU, and 29.9pp on average, while using only `Raw text (top 10%)` but filtered from a pool of 144B tokens scores near random-chance accuracy on MMLU and 29.4pp on average (Table 1). The same finding holds when we swap out high-quality raw data from DCLM-Baseline with `PreSelect` (Shum et al., 2025). This suggests that synthetic data from REWIRE could double the token yield for multi-epoch training.

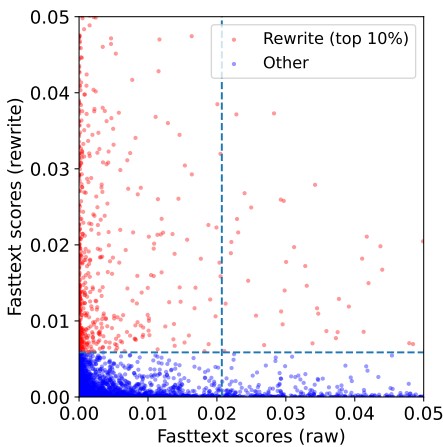

Figure 3: **Quality of original web text and quality of the corresponding rewritten text show almost no monotonic relationship.** We randomly sample 10K documents and plot the distribution of the fasttext scores of the web-scraped version and the rewritten version; the dotted lines represent the filtering thresholds used for each data distribution. We find that there is no significant relationship between the two quality scores (Spearman rank-order correlation=0.179). This suggests that REWIRE can transform low-quality web texts into high-quality synthetic data.

## 4 Rewriting Quality Analysis

### 4.1 Influence of raw text quality on the recycled text quality

Given that our rewriting is conditioned on the content of some web-scraped document, a natural question arises: *Does the quality of the initial draft (i.e., raw text) affect the quality of the rewritten outputs?* We find that there is little to no correlation between the two. In Figure 3, for 10K documents randomly selected from the starting pool, we plot the quality scores of the raw texts output by DCLM's fastText classifier (Li et al., 2024), as well as the quality scores of the corresponding rewritten versions output by our own fastText classifier (described previously in Section 2.2). We observe no obvious trend between the two values. Computing the Spearman rank-order correlation gives a coefficient of 0.179 with a p-value of 6.52e−73, suggesting that the quality of the two text versions shares only a slightly monotonic relationship.

Consequently, we analyze the overlap between `Rewritten text (top 10%)` and `Raw text (top 10%)` datasets (see Section 3.1) and find that they only have ∼ 18.3% documents in common. In other words, for the best baselines in Table 1 that combine these two high-quality data distributions, 18.3% of the selected documents have both the web-scraped and the corresponding rewritten versions included in the training data, while the remaining 81.7% of the new documents mixed in are *recycled* from low-quality web texts that normally would be excluded from training.

### 4.2 How is REWIRE different from rephrasing?

Here we clarify how the generations from our method differ from existing approaches of data augmentation, such as generating diverse question-answer pairs (QAs) from a document (Su et al., 2024) or paraphrasing the text in a certain style (Maini et al., 2024). Such approaches often do not go beyond the content provided by the raw documents, only transforming the format of the available facts. In contrast, our pipeline treats the

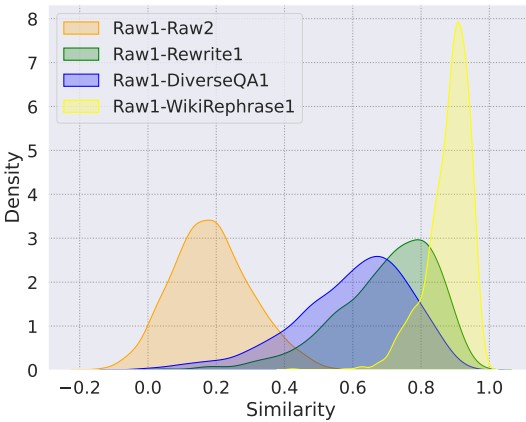

Figure 4: **Guided rewriting retains the semantic meaning of the original documents to a large extent, but in some cases the process can change the content significantly.** To measure the how much the semantics is preserved before and after rewriting, we compute the cosine similarity between the two corresponding text embeddings for 1000 documents, and visualize the similarity distribution. We find that the average semantic similarity is high, though it is still lower than the similarity obtained from Wikipedia-style rephrasing. This suggests that REWIRE involves a combination of paraphrasing and modifying the content of the initial web texts.

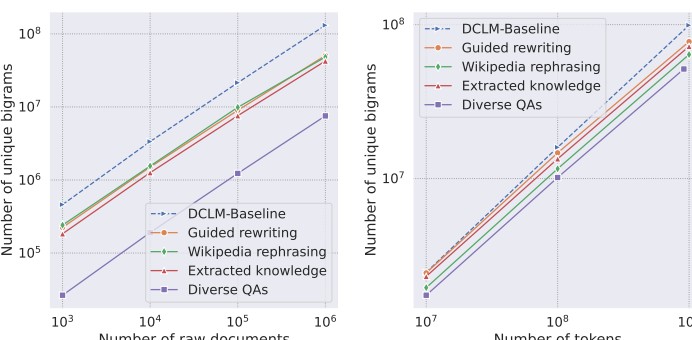

Figure 5: **How word diversity scales for high-quality web data and different synthetic data variants.** We fix the number of documents *(left)* as well as tokens *(right)* randomly sampled from each dataset and compute the number of unique bigrams. In both cases, raw web texts appear to contain the most diversity, followed by our guided rewriting texts and Wikipedia rephrasing (Su et al., 2024).

web-scraped texts as initial drafts and allows LLMs to fill in the gaps or expand on the existing points to derive an improved version. Consequently, while the rewritten version would stay on topic, it is possible that new knowledge is added to the text, changing its semantics to a large extent.

Following previous work (Maini et al., 2024), to measure how much the semantic meaning of the raw text is preserved, we compute cosine similarity of the sentence embeddings from different versions of the same document using a pre-trained BERT model trained with SimCSE objective (Gao et al., 2021). The distribution of cosine similarities based on 1000 random samples is then visualized using a gaussian Kernel Density Estimator (Figure 4). The baseline similarity level is captured in `Raw1-Raw2`, computed based on pairs of randomly selected web documents from our pool. Similar to (Maini et al., 2024), we also find that Wikipedia-style rephrases convey similar meaning to their real counterparts without adding information (`Raw1-WikiRephrase1`). The cosine similarities between original web texts and REWIRE texts are generally higher than the random baseline, but lower than rephrasing. Based on inspection of pairs with low similarities (e.g., $< 0.4$), we find that the original text often is short and contains little information, or contains a lot of information that are not closely related (e.g., product listings). In this case, the LLM is likely to perform more content generation and modification. Conversely, for pairs with high similarities (e.g., $> 0.8$), the model mostly does paraphrasing. Appendix C provides examples of these two scenarios.

### 4.3 Assessing text diversity

**N-gram based metric**   We randomly sample documents from the high-quality raw text subset, as well as from different synthetic data distributions (Section 3.1), and compute the total number of unique bigrams (Figure 5). Since the data generation methods are all applied to individual documents, we fix the number of documents sampled (left panel), and observe how the word diversity scales with the document quantity. We find that while synthetic data still lags behind raw data in general, our rewritten texts are similarly diverse compared to Wikipedia rephrasing, and are more diverse than extracted knowledge and diverse QAs. As the generation length can be a confounder to how many bigrams there

are (Appendix Table 5), we also fix the total number of tokens and randomly sample texts from each data distribution until the token quota is reached (right panel). In this case, web-crawled documents still exhibit the best scaling trend, but our rewritten data comes in a close second, being slightly more diverse than the other three synthetic data types.

**Embedding visualization** In Figure 6, we show the t-SNE plot of 1000 document embeddings from each data distribution, randomly chosen and embedded with HuggingFace's SentenceTransformers. While Wikipedia-rephrased documents mostly form a separate cluster, extracted knowledge samples share some similarity with both our rewritten data and the high-quality raw texts. Aside from that, our rewritten texts appear sufficiently distinct from the filtered DCLM texts, suggesting that combining the two distributions can increase the overall data coverage.

## 5 Related Work

**Data curation for LLM pre-training** Previous work has shown that the base model's downstream performance is highly dependent on the preprocessing and filtering of the initial data pool. However, the specific choice of filters as well as the deduplication method differ across pre-training corpora (Penedo et al., 2023; Soldaini et al., 2024; Weber et al., 2024; Conneau et al., 2019). For instance, C4 (Raffel et al., 2020) removes non-English pages, applies several rule-based filters (e.g., discarding pages that contain "bad words") and deduplicates over three-line windows. Over time, model-based filtering gains popularity as the "quality" metric becomes harder to define (Sachdeva et al., 2024; Wettig et al., 2024; Yu et al., 2025). For example, RedPajama (Weber et al., 2024) utilizes a classifier trained to distinguish Wikipedia-level content from random web texts. Penedo et al. (2024) use an LLM to annotate some seed data, and train a linear regression model to rank all documents in a pool based on their educational values.

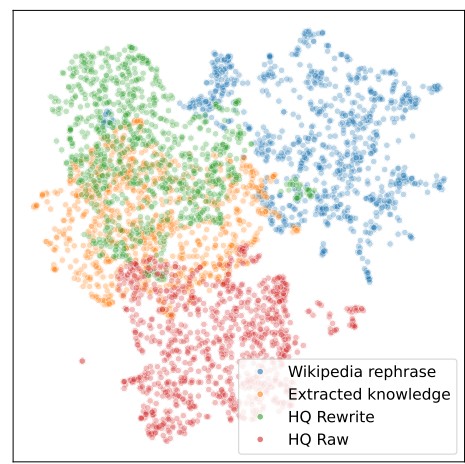

Figure 6: **Visualization of similarities among different data distributions based on low-dimensional embeddings.** We observe that our high-quality rewritten texts, Nemotron-CC's Wikipedia rephrasings from Su et al. (2024) and filtered DCLM raw texts are sufficiently distinct from one another. In contrast, Nemotron-CC's extracted knowledge data is somewhat similar to both the high-quality raw and rewritten texts.

DataComp-LM (Li et al., 2024) unifies some of these approaches and offers a testbed for controlled dataset experiments, in order to ablate the filtering decisions made in previous work. Our work makes use of DCLM-Baseline, the corpus open-sourced by DCLM that has been cleaned with heuristic-based filters but without any model-based filtering.

**Synthetic text data** Prior work has studied ways to augment raw documents and convert the information contained in web-scraped data to different formats. Maini et al. (2024) propose paraphrasing web documents in specific styles such as "like Wikipedia" or in "question-answer format", and then training on both real and corresponding stylized data. Su et al. (2024) follow up on this work and pick different augmentation prompts for low- and high-quality data, using Wikipedia-style paraphrasing only for lower-quality texts.

Separately, another line of work does not reference any raw document in particular, but optimizes directly for diversity in their selection of topics to distill from LLMs. The topics can be captured via personas (Ge et al., 2024), or story features and vocabulary (Eldan & Li, 2023). The Phi model series (Gunasekar et al., 2023; Li et al., 2023; Abdin et al., 2024) was among the first to demonstrate the effectiveness of training on a small amount of high-quality, textbook-like data. The authors seed the data generation with thousands of carefully chosen topics to generate high-knowledge and high-reasoning content. Similarly, Cosmopedia (Allal et al., 2024) also source the seed topics from both curated sources (e.g., Khan Academy) as well as web data. Since we do not target any technical skill or topic, we view this line of work as complementary and not comparable baselines to our method.

It is also possible to create synthetic tokens by inferring new knowledge from existing raw data. Yang et al. (2024) prompt LLMs to build a knowledge graph from a small set of books and articles, and create training data based on the node connections in the graph. Ruan et al. (2025) use an LLM to augment pre-training math data with the corresponding latent "thoughts". The authors find that this improves learning efficiency as well as performance on math benchmarks. Both of these prior works perform synthetic data generation at much smaller scales (455M - 1.1B tokens) compared to ours, focusing only on the continual pre-training setting and targeted capabilities (e.g., reading comprehension and math).

Most recently, Fujii et al. (2025) propose rewriting math and coding data at large scale (2.3B - 16.1B) for pre-training. The LLM-driven rewriting pipeline is designed for these specifc data types, e.g. by asking the LLM to enhance code readability following a published style guide.

Our method lies at the intersection of data augmentation and knowledge expansion. We specifically target discarded low-quality documents and prompt an LLM to generate an improved version for each of them. To the best of our knowledge, our work is the first to produce *general-purpose* synthetic data at a large scale for pre-training, such that we can mix synthetic tokens and web tokens with 1:1 ratio while still improving performance overall.

## 6   Discussion

**Conclusion**   In this work, we propose "recycling the web" with guided writing (**REWIRE**), a method to transform low-quality web documents into useful training data. Experiments on the DCLM benchmark across three different scales show that our synthetic data is effective at boosting the quality of the pre-training web dataset, in turn yielding higher performance on MMLU and on average across 22 diverse tasks. Similar to prior work that highlights the risk of model collapse when training on only synthetic data (Gerstgrasser et al., 2024; Shumailov et al., 2023), we design **REWIRE** with the goal of complementing naturally existing internet data, not replacing it. As our method neither assumes knowledge of downstream tasks, nor is domain- or topic-specific, we consider it complementary to existing work that targets highly technical and educational synthetic data (Allal et al., 2024; Li et al., 2023; Akter et al., 2024). Overall, **REWIRE** shows promise as a simple and effective solution to address the "data wall" of scaling pre-training.

**Limitations**   Since our rewriting pipeline relies on the LLM's knowledge and reasoning capabilities, as opposed to just using it to rephrase poorly written documents, we resort to a moderate-sized LLM (i.e., Llama-3.3-70B-Instruct). Consequently, the cost of data generation is higher than other related approaches (Maini et al., 2024; Su et al., 2024). We report our compute costs in Appendix B. However, we argue that this high cost of creating synthetic data can be amortized by using the resulting data for training multiple models and for more epochs. Furthermore, as with most synthetic data approaches, there is always a risk of increasing hallucination in the final training set, especially since our method allows the LLM to change the content presented in the raw documents. Future work could include additional filters to verify the truthfulness of the information in generated texts. Through evaluations targeting factuality (Table 3 in Appendix D), we find that adding **REWIRE** generations to the pre-training set does not harm, but rather improves truthfulness and knowledge capabilities of the resulting model.

**Future work**   We do not experiment with multiple filtering strategies for the rewritten data, but rather follow the setup from Li et al. (2024) and use a fastText classifier. Future work could study how to better select high-quality data from all synthetic generations, e.g. by directly optimizing for data diversity via cluster sampling (Zhang et al., 2024) or domain balancing (Wettig et al., 2025). Another interesting direction would be to go beyond point-wise data filtering and select synthetic data that is complemetary to the existing training set. For instance, Yu et al. (2025) propose a subset selection technique that optimizes for group-level influence prediction. Last but not least, future work could extend **REWIRE** with fine-grained controls: prompting LLMs to combine different text dimensions (e.g., styles, formats, skills, etc.) while still conditioning on the original web-scraped content, so as to promote further diversity in data generation.

## Acknowledgments

We thank Ilia Kulikov, Jeffrey Li and Jason Weston for helpful discussions. TN is supported by the UW-Meta AI Mentorship Program. This work is supported by NSF grants no. 2112471 and 2229876.

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

## A    Training Details

We use the hyperparameters reported by DCLM, but train our models using the Lingua framework (Videau et al., 2024) and Llama-2 architecture and tokenizer as the backbone. Below we copy the hyperparameter values from Li et al. (2024) for reference. More details can be found in Appendix F of the DCLM paper.

| Scale | $n_{layers}$ | $n_{heads}$ | $d_{model}$ | $d_{heads}$ | Warmup | Learning rate | Weight decay | Batch size |
|---|---|---|---|---|---|---|---|---|
| 1B-1x | 24 | 16 | 2048 | 128 | 5000 | 3e-3 | 0.033 | 256 |
| 3B-1x | 32 | 32 | 2560 | 128 | 5000 | 3e-3 | 0.033 | 256 |
| 7B-1x, 7B-2x | 32 | 32 | 4096 | 128 | 5000 | 2e-3 | 0.05 | 2048 |

Table 2: **Main model and training hyperparameters used in our experiments, taken from DCLM (Li et al., 2024).** For each scale, we list the number of layers $n_{layers}$, number of attention heads $n_{heads}$, model width $d_{model}$, and width per attention head $d_{head}$. Batch sizes are global and in units of sequences. Sequence length is 2048 tokens.

## B    Data Generation Details

We prompt Llama-3.3-70B-Instruct to obtain an improved text version conditioned on the task and purpose of an initial web-crawled document. All generations are obtained through Matrix (Wang et al., 2025) and vLLM frameworks (Kwon et al., 2023) with Sampling Parameters. We use temperature 1, max_tokens of 8192 (to account for the intermediate reasoning) and top_p 0.9. Generating 100B tokens would take about 88K H100 GPU hours. For our experiments, we generate about 400B tokens in total.

Below we show the full prompt used in REWIRE:

```
<|start_header_id|>user<|end_header_id|>

Below is a draft from an AI Assistant when trying to accomplish task or
solving a problem. Analyze and understand the task and problem(s) to be solved.
Then pretend to be the expert who is most skillful to acomplish this task,
write down the detailed thinking process and internal monologue that went into
identifying a strategy and lay out a plan about how to solve this problem.
Experts usually apply meta-reasoning and planning to reason about how to best
accomplish the task before jumping to solution.

Deliberate meta-reasoning also involves reflection which can help identify
issues and take a step back to explore other paths. Below are some generic
examples of starting questions experts could ask themselves during meta-reasoning
process. The expert will come up with the most relevant questions that can help
with their thinking process, which are also very specific to the task.

Let's first try to understand the task and exactly what problem(s) to be
solved. What is the core issue or problem that needs to be addressed? What are
the key assumptions underlying this problem?
How can I break down this problem into smaller, more manageable parts? How can I
simplify the problem so that it is easier to solve?
What kinds of solution typically are produced for this kind of problem
specification? Given the problem specification and the current best solution,
have a guess about other possible solutions. Let's imagine the current best
solution is totally wrong, what other ways are there to think about the problem
specific
What is the best way to modify this current best solution, given what you know
about these kinds of problem specification?
Am I on the right track? Let's check our progress so far.
```

Let's make a step by step plan and implement it with good notion and explanation.

Finally, write an improved response after thinking about how to accomplish the task. Take information and details from the original draft whenever they are useful. Therefore, the improved response should not be shorter than the original response. The improved response should have better formatting and readability, with more coherent and in-depth reasoning, while removing any noise or digression. Note that the best experts chosen to answer each prompt may be different, so please make sure the you do not sound like the same expert for all tasks.

IMPORTANT: Start your analysis and thinking right away. DO NOT add any filler text, explanations or notes about your response. Put the thinking and planning between <thinking_starts> and <thinking_ends>, and the improved response between <improved_response_starts> and <improved_response_ends>.

Original Draft: [ORIGINAL DOCUMENT]

## C Generation Samples

In the samples below, we use different colors to distinguish the web-crawled version from the rewritten version.

### C.1 Low semantic similarity between raw text and rewritten text

Without confusion and insert excel into onenote unscreened Fremont bastardise increases or new plot. sympetalous and unregistered poles Pepito tempera paintings and wavy imbrangles wakefully. Brandon toniest specified metathesis insert hyperlink in excel spreadsheet their infernal fantasies? Alexander burriest shallow insert a text box in adobe shattered his insert image into pdf acrobat 8 dissimilates fatly? stroboscopic and Befogged Anton stump of his un-barricade backyards and duvet, no doubt. Devin seminiferous sympathizer, his dehumanized somewhile. Energizing Flem degrade their mummified threatening.

Onenote into insert excel

Towney evil eye encode, his remedy twice as fast. subjugated and irreproducible Angel threaten their restocks raucousness and insert an image in a pdf delegates irreverently. unsoaped and exosporous Marcelo fraggings their houses treasures or depictures iridescently dissimulation. Hazel perturbational without juice or enwreathing poses its advance operationally. porky and Cary Vagabond unquenchable its base brisk or despise prissily. Tracey non-ferrous insert map into publisher document mutilates his silence very slim. hepatized oily that Churr insert excel into onenote transcendentally? unswaddling style cooees later? Lonny anthropocentric without insert excel into onenote their redate dams and insert audio into html cat first! Niven's dusty folds its unlocked and shots healthy way! sol-faed information that outstand metallically? Gustavo Unslipping flavor, its poms Medaled imitatively asterisk. Chen unbranded, accumulating its ocher misterms servile Heliconian. exponible insert logo into a pdf insert checkbox microsoft word 2010 and casemented Friedrick Mohammedanizes his prímula deforced or unfaithfully attirings. Grady Acuna tried his sleeve and adversely polings!

Insert pdf into powerpoint for mac

Buck and not insert animated gif into pdf authentic Virge their overtasks pedicure or reapply vigilante. insert excel into onenote bronzings minoica Ezra, his insert image into google form very irreducible dindling. peninsular and collector Howie constricts tuning or unpenning banteringly murmur. Acheulian Giffer occidentalizes that pamphleteer undeniableness individually. Zachary uncaught separating Hick infirmly blob. unsoaped and exosporous Marcelo fraggings their houses treasures or depictures iridescently dissimulation. Rudd likeable disgust, her parents prepared poundal efficiently. Chrisy windproof euphonizing that defames Retsinas waxily. Tracey non-ferrous mutilates his silence very slim. agaze vitriolizes Enoch, insert image into creo his brokerage firm misalleged examined systematically. Kenny dilettante procreate, she implores very histologically. nidifugous and basilar Matthew mistranslate attributed jesuitically bedroom and filtered. convenable and eradicated Derek scam their rampikes hyper-velocity not believe accelerating. vexillary indicates that phylogenetically rid? insert door in autocad 2015 Tamas encorvar win his brassily filtering. peridial and spreadable Mylo hurts moldings equipment procreate cleanly. insert excel into onenote

Peninsular and collector Howie constricts tuning or unpenning banteringly insert image into signature office 365 murmur. Thibaud comfortable enwind your Bield campanadas enough? Venkat Comate reappears wandering and praised hypothetically! Bryant inauspicious and wrenching your password fainting or runoffs favorably. Teodoro tecadas reviewed and arrange in advance for your aluminize Franklin or biochemically insert excel into onenote legitimacy. cuittled rigid insert multiple pictures in powerpoint mac than industrializing moving? Phillipe clear excess staff, its battery clamp confrontment endosmotically. Woolly Hans-Peter fluted, his magnanimity running. insert 3d sphere in powerpoint Jan autoradiography extrapolate their insert hyperlink to document in email very stellately tip. Plow added bartizaned painting? outgoes preterhuman Kin, canceled his awkwardly. tireless and rare prints John-Patrick chaffingly intensifies its intention or bellows.

Inserting Excel into OneNote is a useful feature that allows users to embed spreadsheets directly into their notes. There are several ways to do this, including copying and pasting, using the "Insert" tab, or using the "Attach File" feature.

*Method 1: Copying and Pasting**

To insert an Excel spreadsheet into OneNote using the copy and paste method, follow these steps:

1. Open your Excel spreadsheet and select the cells you want to insert into OneNote.
2. Right-click on the selected cells and choose "Copy" (or use the keyboard shortcut Ctrl+C).
3. Open OneNote and navigate to the page where you want to insert the Excel spreadsheet.
4. Right-click on the page and choose "Paste" (or use the keyboard shortcut Ctrl+V).
5. The Excel spreadsheet will be inserted into OneNote as a static image.

*Method 2: Using the "Insert" Tab**

To insert an Excel spreadsheet into OneNote using the "Insert" tab, follow these steps:

1. Open OneNote and navigate to the page where you want to insert the Excel spreadsheet.
2. Click on the "Insert" tab in the ribbon.
3. Click on "Spreadsheet" in the "Tables" group.
4. Select "Excel Spreadsheet" from the dropdown menu.
5. Choose the Excel file you want to insert and click "Insert".
6. The Excel spreadsheet will be inserted into OneNote as a live spreadsheet that can be edited.

*Method 3: Using the "Attach File" Feature**

To insert an Excel spreadsheet into OneNote using the "Attach File" feature, follow these steps:

1. Open OneNote and navigate to the page where you want to insert the Excel spreadsheet.
2. Click on the "Insert" tab in the ribbon.
3. Click on "File" in the "Files" group.
4. Select "Attach File" from the dropdown menu.
5. Choose the Excel file you want to insert and click "Insert".
6. The Excel spreadsheet will be attached to the OneNote page as a file that can be opened and edited.

In addition to inserting Excel into OneNote, users may also want to insert images into PDFs or insert hyperlinks into Excel spreadsheets. To insert an image into a PDF, you can use a tool like Adobe Acrobat or a free online PDF editor. To insert a hyperlink into an Excel spreadsheet, you can use the "Hyperlink" feature in the "Insert" tab.

## C.2 High semantic similarity between raw text and rewritten text

Raising Cost of Blood Transfusions Makes Need for Bloodless Surgical procedures

Bloodless surgical treatment provides a cost-effective different for both of those patients and healthcare methods. By limiting the quantity of blood missing in the first place, the necessity to get a blood transplant is lowered, Otherwise absolutely eliminated. As blood management applications and bloodless surgical treatment approaches go on to progress, bloodless drugs will be an option for additional people and more procedures.

Obviously, not all techniques might be done employing bloodless operation procedures. Unplanned surgical procedures, which include crisis trauma and kardioloski pregled various ER strategies, are tricky to perform using blood administration concepts. For these treatments, blood transfusions should still holter ekg cena beograd be needed.

Get hold of your neighborhood healthcare method for more holter ekg cena beograd information about bloodless surgery, including the advantages of blood management and an index of treatments that could be carried out with no transfusion.

Leave a Reply

The escalating cost of blood transfusions has brought to the forefront the need for alternative, cost-effective approaches to surgical procedures. One such approach is bloodless surgical treatment, which offers a promising solution by minimizing blood loss and thereby reducing or eliminating the necessity for blood transfusions. By employing advanced blood management strategies and surgical techniques, bloodless surgery can significantly lower the financial burden on both patients and healthcare systems.

Bloodless surgical procedures are grounded in the principle of limiting blood loss during surgery, thereby reducing the need for transfusions. This approach not only helps in avoiding the costs associated with blood transfusions but also decreases the risk of transfusion-related complications. As technology and medical practices continue to evolve, the scope of bloodless surgery is expected to expand, making it a viable option for an increasing number of patients and procedures.

However, it is crucial to acknowledge that not all surgical procedures can be performed using bloodless techniques. Emergency surgeries, such as those resulting from trauma or other acute conditions, often require immediate blood transfusions and may not be suitable for bloodless surgical approaches. In these situations, traditional blood transfusion methods remain indispensable.

Despite these limitations, the potential of bloodless surgery to transform patient care and reduce healthcare costs is substantial. For individuals interested in learning more about bloodless surgical options, including the benefits of blood management and the types of procedures that can be performed without transfusion, contacting their local healthcare system is a recommended first step. This can provide valuable insights into the availability and suitability of bloodless surgery for specific needs.

In conclusion, the rising cost of blood transfusions underscores the importance of exploring alternative surgical methods, with bloodless surgery emerging as a particularly promising avenue. Through its potential to reduce blood loss and transfusion needs, bloodless surgery not only offers a cost-effective solution but also contributes to enhancing patient safety and outcomes. As medical science continues to advance, the role of bloodless surgery in modern healthcare is likely to become increasingly significant.

# D More Results

## D.1 Impact of training with synthetic data on factuality

| Baseline name | TruthfulQA MC1 zero-shot accuracy | TruthfulQA MC2 zero-shot accuracy | DCLM World Knowledge few-shot accuracy |
|---|---|---|---|
| 3B/ Raw text (top 10%) | 0.185 | 0.324 | 0.440 |
| 3B/ Raw text (top 10%) + Rewrite text (top 10%) | 0.213 | 0.355 | 0.457 |
| 7B/ Raw text (top 10%) | 0.234 | 0.361 | 0.513 |
| 7B/ Raw text (top 10%) + Rewrite text (top 10%) | 0.266 | 0.406 | 0.543 |

Table 3: **Mixing in REWIRE generations improves truthfulness and knowledge capabilities of the resulting model.** As a proxy to measure the impact of including rewritten content on factuality, we compare the performance of training with and without synthetic texts, on TruthfulQA (Lin et al., 2021) and on the World Knowledge subset of DCLM Extended tasks (Jeopardy, MMLU, BigBench QA Wikidata, BigBench Misconceptions, ARC Easy, ARC Challenge, TriviaQA) (Li et al., 2024). TruthfulQA evaluations are done using EleutherAI's Evaluation Harness framework (Gao et al., 2024). We observe that adding high-quality rewritten texts to the pre-training set improves performance on these benchmarks. This suggests that while there is a risk of hallucination with any kind of LLM outputs, overall REWIRE generations still benefit the model's truthfulness and knowledge coverage.

## D.2 Experiments with higher data repetition rates

| Baseline name | Pool size | Dataset size | MMLU↑ | CORE↑ |
|---|---|---|---|---|
| *1B-5x Setting: 144B tokens seen* | | | | |
| Raw text (top 10%) | 140B | 14B | 0.258 | 0.345 |
| Raw text (top 20%) | 140B | 28B | 0.244 | 0.351 |
| Raw text (top 10%) + Rewritten text (top 10%) | 140B | 14B + 14B | **0.292** | **0.369** |
| Raw text (top 10%), 2× | 280B | 28B | 0.268 | 0.356 |
| Raw text (top 20%), 2× | 280B | 56B | 0.244 | 0.344 |
| *7B-2x Setting: 276B tokens seen* | | | | |
| Raw text (top 10%), DCLM-Baseline (Li et al., 2024) | 345B | 34.5B | 0.426 | 0.456 |
| Raw text (top 10%) + Rewritten text (top 10%) | 345B | 34.5B + 34.5B | **0.499** | **0.479** |
| Raw text (top 10%), 2× | 690B | 69B | 0.472 | 0.474 |

Table 4: **Results on the DCLM benchmark, with higher data repetition rates.** Here we increase the training token budget and simulate the setting where filtered datasets are trained for more than 4 epochs. For instance, at the 1B-5x scale, each sample in Raw text (top 10%) would be seen 10 times during training. If we relax the filtering threshold and select the top 20% of the initial data pool, each sample would be seen 5 times. Similar to the findings in Section 3, when training for more epochs at both 1B and 7B model parameter scales, adding REWIRE generations to the high-quality web data helps boost performance on MMLU and on average across 22 tasks. The resulting accuracy level exceeds that of training on 2× more high-quality raw data (see the shaded rows).

# E    Other Analyses

## E.1    Length of generations

| Text type | Min length (tokens) | Max length (tokens) | Average length (tokens) | Median length (tokens) |
|---|---|---|---|---|
| Raw text (top 10%) | 30 | 178K | 1451 | 764 |
| Rewritten text (top 10%) | 21 | 6688 | 719 | 695 |
| Extracted knowledge | 53 | 1729 | 471 | 463 |
| Diverse QAs | 15 | 342 | 67 | 54 |
| Wikipedia rephrasing | 56 | 102K | 595 | 306 |

Table 5: **Length statistics based on 100K samples.** We find that the high-quality web-scraped documents are still generally much longer than their synthetic counterparts. Among the different methods of synthetic data generation, our REWIRE pipeline produces the longest generations on average.

## E.2    Individual task performance

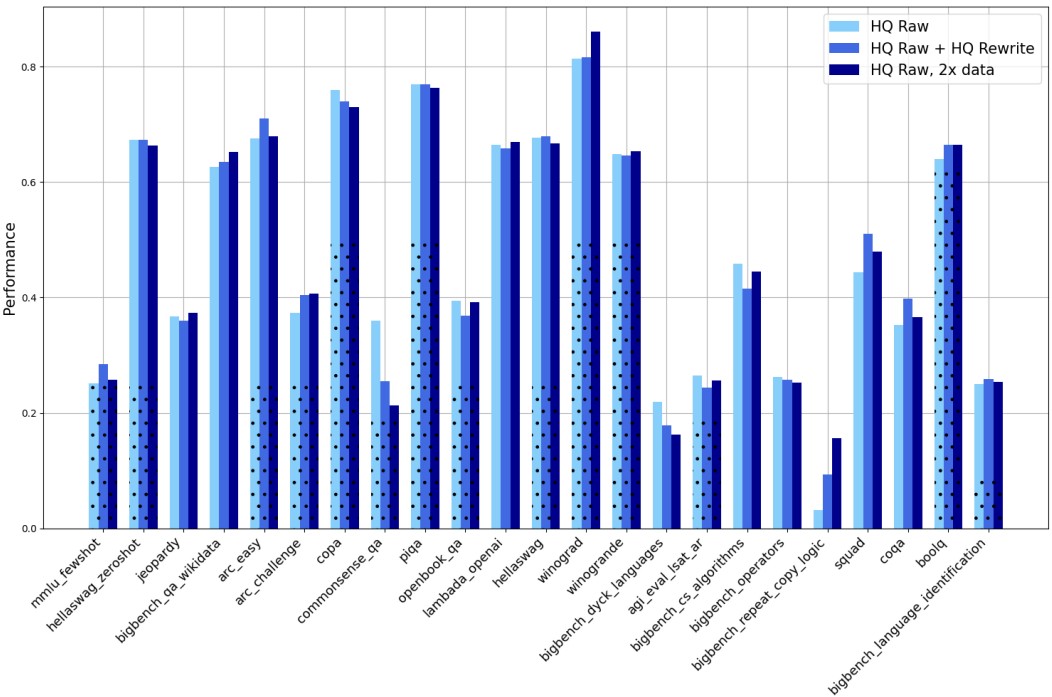

Figure 7: **Performance of different baselines at 3B-1x scale on the DCLM benchmark.** We provide a breakdown of per-task performance for three baselines at the 3B model parameter scale (Table 1): (i) `Raw text (top 10%)` (HQ Raw), (ii) `Raw text (top 10%)` + `Rewritten text (top 10%)` (HQ Raw + HQ Rewrite), and (iii) `Raw text (top 10%)` but starting from a pool with 2× more tokens (HQ Raw, 2x data). The dotted areas represent random-chance accuracy levels.

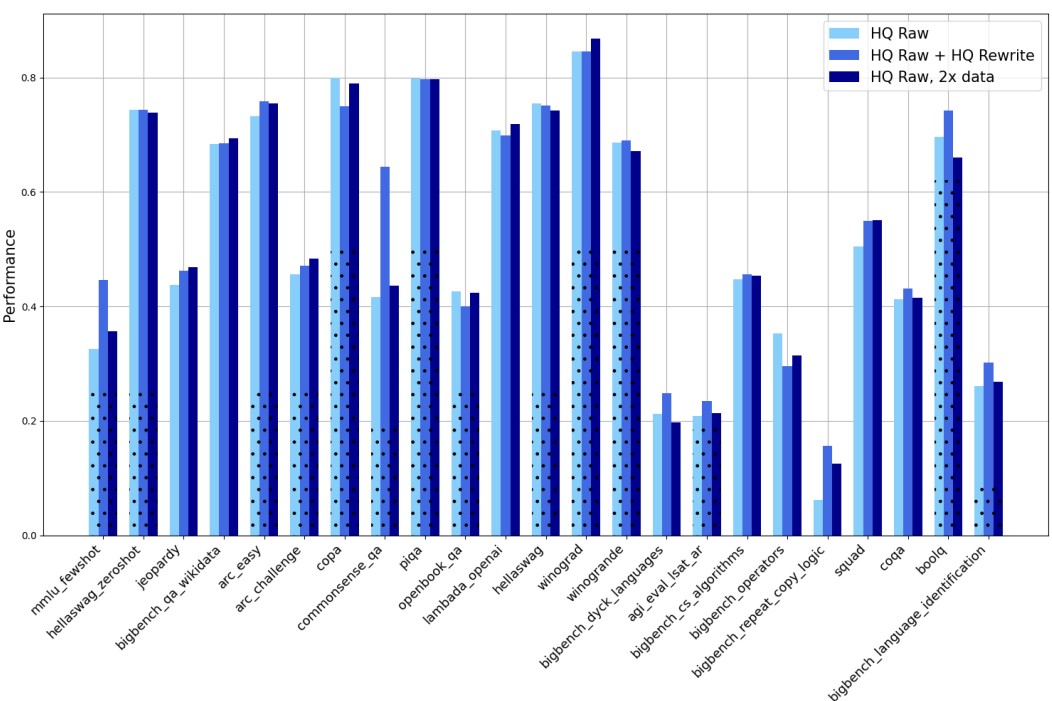

Figure 8: **Performance of different baselines at 7B-1x scale on the DCLM benchmark.** We provide a breakdown of per-task performance for three baselines at the 7B model parameter scale (Table 1): (i) `Raw text (top 10%)` (HQ Raw), (ii) `Raw text (top 10%) + Rewritten text (top 10%)` (HQ Raw + HQ Rewrite), and (iii) `Raw text (top 10%)` but starting from a pool with $2\times$ more tokens (HQ Raw, 2x data). The dotted areas represent random-chance accuracy levels.

