# OpenReview forum: "Recycling the Web: A Method to Enhance Pre-training Data Quality and Quantity for Language Models"
_colmweb.org/COLM/2025/Conference — COLM 2025_

### Official Review · Reviewer_VaHx · 2025-05-07

**Rating:** 6
**Confidence:** 3
**Ethics Flag:** 1

**Summary:**

This work proposes a method that recycles filtered-out, moderate-quality documents by prompting an LLM to rewrite them into higher-quality versions. The motivation is to leverage the abundant information present in moderate-quality data, which is typically discarded during data curation. Experimental results show that supplementing high-quality pretraining data with these rewritten samples can improve downstream performance of trained language models.

**Reasons To Accept:**

1. The paper addresses an underexplored yet practical data curation strategy—reusing discarded moderate-quality data via LLM rewriting—which is both novel and intuitive.

2. The authors conduct extensive experiments using two model scales and datasets from two different sources. They also include detailed ablation studies across data scales to mitigate concerns about robustness.

**Reasons To Reject:**

1. Although the authors mention a focus on long-context modeling (line 79), there is no corresponding experimental setup or analysis to support this claim.

2. The paper presents only average performance across tasks. It would be more informative to include per-task results to assess whether the improvements are consistent or confined to a subset of task types.

**Minor Comments:** Figure 1 is unclear.

---

> ### Author Response · Authors · 2025-06-02
> **Response to Reviewer VaHx**
>
> We thank the reviewer for the helpful feedback!
>
> **Long-context modeling.** To clarify, we did not say that our work focuses on long-context modeling. In line 79, we describe our setup as in line with “long token horizon training”, which is different from long-context modeling. Long token horizon training (see [1]) involves a relatively large token budget (compared to the number of high-quality data available for training) and thus necessitates making multiple passes over the training set. We simulate this setup by fixing the total number of tokens curated with different baselines and training on the resulting datasets for 2-4 epochs.
>
>
> **Include per-task results.** We have included an analysis of per-task performance difference at the 3B model scale in Appendix D.2 Figure 6. At this scale, mixing raw and synthetic tokens significantly improves over using only the former on MMLU, ARC Easy, ARC Challenge, SQUAD, COQA, BoolQ and some BigBench tasks (Repeat copy logic + language identification).
>
> Are the performance improvement consistent? We have generated more synthetic tokens with REWIRE and obtained additional results at the 7B model scale. To summarize, the same observations hold: (i) compared to training on only the highest-quality raw data, mixing it with our high-quality rewritten texts leads to 12% improvement on MMLU and 2.5% improvement on average across 22 tasks of DCLM’s CORE, (ii) training on the raw & synthetic mix is also more effective than having access to 2X more web data. Refer to the Global Response for more details.
>
>
> **Figure 1 is unclear.** We will try to improve the visualization in the next iteration of the paper.
>
> [1] Su, Dan, et al. "Nemotron-CC: Transforming Common Crawl into a Refined Long-Horizon Pretraining Dataset." arXiv preprint arXiv:2412.02595 (2024).

---

> > ### Comment · Reviewer_VaHx · 2025-06-08
> > **Response to Author Rebuttal​**
> >
> > Thanks for the author's response, which addresses my concern. I remain positive about this.

---

> > > ### Author Response · Authors · 2025-06-10
> > > **Author response**
> > >
> > > Thank you for acknowledging our rebuttal! We are glad to hear that our response has addressed your concerns.

---

> ### Comment · Reviewer_Mvmh · 2025-06-08
> **Reviewer response**
>
> Thank you for the response. I would like keep my scores the same.

---

### Official Review · Reviewer_NFsi · 2025-05-11

**Rating:** 5
**Confidence:** 4
**Ethics Flag:** 1

**Summary:**

Pre-training corpus for language model contains many low-quality content. This paper presents a way of using LLMs to rewrite these low-quality documents so that they could be useful for pre-training. Experiments on smaller scaled DCLM (1B and 3B) settings showcase the effectiveness of the proposed rewriting method, leading to an average of 1.0 and 1.3 across downstream tasks. The paper also presents comparison of the proposed way of rewriting against other synthetic data generation pipelines, including Wikipedia style paraphrasing, synthesizing QA pairs (Nemotron-CC), and knowledge extraction (Nemotron-CC).

The core of the method is "using an LLM to identify the purpose of the text content of a document, and then asking the LLM to come up with an improved document conditioned on chain-of-thought reasoning". Although the paper is clearly written and the experiments are rigorous, the missing baselines / discussions really limits the novelty of this work. Further comparisons and analysis of the effect of data synthesizing on the effect of both semantic and syntax diversity of data is needed.

**Questions To Authors:**

Comments:

1. Figure 1 is not in SVG format (pdf). I would suggest to use SVG format to enhance the readability of the figure. Also it would be better to make annotations on which part of the pipeline does rewire contributes the most (rewriting documents).

2. Can you show the cost of rewriting the documents in terms of # of GPU hours?

**Reasons To Accept:**

1. The paper studies a very important and timely topic, as high-quality data is soon becoming consumed, the need for synthesizing high-quality data, as well as transforming mid-to-low quality to high quality data is needed.

2. The paper's experiment setting is rigorous, containing detailed description and many different baselines. The results are able to outperform the strong DCLM baseline.

3. The writing is clear, concise, and contains discussions on the effect of rewriting web text on the diversity of text.

**Reasons To Reject:**

1. Important discussion / baselines missing: Both "Synthetic Continued Pre-training" [1] and "ProX" [2] uses an LLM to re-write pre-training data examples. Since these are highly similar approaches to the submission, discussions, if not comparisons are needed for the community to understand the strengths and limitation of these similar approaches. Since these two papers are released September 2024, I wouldn't consider the two as "concurrent work".

2. Related to 1. Since there are many works that consider prompting LM to rewrite (pre-)training data, this really limits the novelty of this work. I acknowledge that the authors differentiate themselves as "our pipeline treats the web-scraped texts as the initial drafts and allows LLMs to fill in the gaps or expand on the existing points to derive an improved version" by showing low semantic overlap between original text and rewritten text. It would help to see a quantitative comparison of # of documents with knowledge gaps and rewriting it fills in the knowledge gap.

3. Knowledge in web text are usually written in many different forms: for example, the same knowledge can exist on the web in forms of books, formal language, conversation, question answering pairs, and even songs and poetry. The rich form of natural language make LMs learn knowledge more robustly. However, by rewriting the text, it seems that we are limiting the structure. It would help to see how re-writing affects the structural diversity aside from semantic diversity, and how structural diversity relates to language model generalization. Lacking analysis on structural diversity limits the contribution of this work.

3. The performance gains are fairly limited (<2 points on average). Admittedly, the DCLM baseline is a strong baseline so if the cost of the method is low, the limited gains can be justified. However, the authors did not conduct study on the cost of their method. If the cost is high, then spending a high cost for a limited gain can be hard for people to use this method.

[1] Synthetic Continued Pre-training (Yang et al., ICLR 2025)

[2] Programming Every Example: Lifting Pre-training Data Quality Like Experts at Scale (Zhou et al., 2024)

---

> ### Author Response · Authors · 2025-06-02
> **Response to Reviewer NFsi**
>
> Thank you for your time and for the review!
>
> **Comparison to prior work.** To clarify, our contribution lies in providing a *general-purpose* method to make better use of low-quality web documents that would normally be discarded by SOTA filters. This is especially pertinent in the context of internet data soon being relatively much less available compared to compute resources. Given this motivation, we view the papers you suggested as related but not “highly similar” to our work:
>
> We have discussed "Synthetic Continued Pre-training" [1] in Related Work. [1] synthesizes new knowledge from books, which is more in line with the technical data synthesis direction (e.g. Phi [2]). In contrast, our work doesn’t particularly focus on any type of content. Besides, in [1] the data generation was done at a much smaller scale (for continual training setting) - the authors generated a total of 455M tokens, which is just 6% of the pre-training set size in our smallest experiment (and >30x less than what we generated). We will clarify these differences further in the Related Work discussion.
>
> For ProX [3], we were not aware of this work but will add the citation. ProX’s data refinement approach was applied to existing high-quality documents (e.g. DCLM-baseline). We do not find the refinement operations specified in Table 1 of [3], such as removing noisy lines, string normalization, etc., to be applicable or helpful for improving lower-quality documents (i.e. the emphasis of our work). Furthermore, the refinement procedure happens through a set of predetermined, rule-based function calls, while in our pipeline, the enhancement can be more fluid based on the LLM’s reasoning process.
>
> **Given prior work that prompts LLMs to rewrite data, this limits the novelty of the work.** While the high-level idea of rewriting is not new, to the best of our knowledge, our work is the first to (i) generate synthetic data at a large scale for pre-training, allowing us to mix synthetic tokens with web-scraped tokens with 1:1 ratio, (ii) maintain the quality and diversity of the synthetic texts at such a large scale, such that adding in as much synthetic data as web data during pre-training does not harm but rather improves performance of the base model. The latter is non-trivial.
>
> **Analysis of number of documents with knowledge gaps and rewriting it fills in the knowledge gap.** We have provided analysis in the same vein in Figure 3 of our paper. If we consider Wikipedia-style rephrasing as paraphrasing and not modifying the knowledge gap, we find that our pipeline goes beyond rephrasing and changes the knowledge representation in the majority of the raw documents (see the green curve versus the yellow curve).
>
> **Lacking analysis on structural diversity.** We posit that the structure of the generated texts still largely follows the structure of the original documents, e.g. if the web-scraped document contains a poem, then the LLM would deduce that the task is to write a poem and output that correspondingly. We do not have a compute-friendly way to measure the structural diversity of the synthetic texts (besides asking an LLM to annotate all documents), or to assess how structural diversity affects model generalization (besides trying different ways to sample documents using the LLM annotations and pre-training on different data variations from scratch). These questions could be the scope of another paper altogether, see [4] as an example.
>
> **Cost of data generation/ Does the compute cost justify the performance gain?** We have stated the compute costs used for data generation in Appendix B. In the Global Response posted above, we also highlight our **new results at 7B scale with larger performance gains**: mixing carefully filtered raw texts and rewritten texts leads to 12% improvement on MMLU and 2.5% improvement on average across 22 tasks, compared to training on only the former. This shows that the performance improvement from our method increases with scale. We emphasize that this makes the method especially useful in the current LLM scaling landscape, where AI hardware is becoming more efficient and compute is getting cheaper, while the stock of naturally existing internet data is slowly being exhausted. Consequently, it is worth spending more compute on creating new datasets for pre-training.
>
> **Modifications to Figure 1.** Thanks for the suggestions, we will consider these changes in the next iteration of the paper.
>
> [1] Yang, Zitong, et al. "Synthetic continued pretraining." arXiv preprint arXiv:2409.07431 (2024).
>
> [2] Li, Yuanzhi, et al. "Textbooks are all you need ii: phi-1.5 technical report." arXiv preprint arXiv:2309.05463 (2023).
>
> [3] Zhou, Fan, et al. "Programming every example: Lifting pre-training data quality like experts at scale." arXiv preprint arXiv:2409.17115 (2024).
>
> [4] Wettig, Alexander, et al. "Organize the Web: Constructing Domains Enhances Pre-Training Data Curation." arXiv preprint arXiv:2502.10341 (2025).

---

> > ### Comment · Reviewer_NFsi · 2025-06-10
> > **Reviewer Response**
> >
> > Thank you for your response and especially for clarifying the distinction between this work and others related works. I believe that the authors have addressed most of my concerns. I will update my score accordingly.
> >
> > After another read of this paper, I apologize for having a follow-up concern: Since we are using a stronger LM (in this case llama-3.1-70B-instruct) for rewriting data, does the improvements mostly come from distillation effects? Particularly the knowledge gaps, since larger models generally memorize more information and would be able to correct, refine the low quality data.

---

> > > ### Author Response · Authors · 2025-06-11
> > > **Author response**
> > >
> > > Thank you for acknowledging our rebuttal and for adjusting the rating.
> > >
> > > You raised an insightful question! It is true that new knowledge could be transferred from the LLM used for rewriting to the model trained on the corresponding synthetic data. While existing works distill from a strong LLM by seeding the generation topics or personas (e.g. [1, 2]), we utilize the LLM to partly fill in knowledge gaps by conditioning the data generation on facts, tasks and details already present in web documents.
> > >
> > > The improvement is not purely due to distilling from a stronger model as we show that (i) training on filtered web texts + filtered rewritten texts outperforms training on filtered web texts + even more filtered web texts (i.e. more knowledge from the web) --- see our 2x baselines in Table 1, (ii) the performance improvement from using rewritten data increases with model scale (if it was purely distillation, the gains should decrease as the teacher & student models have less capacity gap).
> > >
> > > We hypothesize that the improvements could be attributed to *both* the strong LLM’s knowledge and reasoning capabilities, as well as the diversity of the information available on the internet (that is currently discarded by existing data filters).
> > >
> > > [1] Gunasekar, Suriya, et al. "Textbooks are all you need." arXiv preprint arXiv:2306.11644 (2023).
> > >
> > > [2] Ge, Tao, et al. "Scaling synthetic data creation with 1,000,000,000 personas." arXiv preprint arXiv:2406.20094 (2024).

---

### Official Review · Reviewer_Mvmh · 2025-05-11

**Rating:** 6
**Confidence:** 3
**Ethics Flag:** 1

**Summary:**

LLMs are typically pretrained on vast amounts of data. The paper explores the concept of guided rewriting of low-quality documents typically discarded during this pretraining process. It shows that incorporating rewritten texts into the filtered data yields better performance than using filtered data alone on downstream tasks.

**Questions To Authors:**

Missing citations:

[1] Rewriting Pre-Training Data Boosts LLM Performance in Math and Code

**Reasons To Accept:**

The paper explores an important topic of how to make effective use of texts during pre-training.

The experiments are thorough and conducted under wide variety of settings, which are crucial in understanding the effectiveness of the approach. The analysis sections are also comprehensive.

The approach shows reasonable results.

**Reasons To Reject:**

From Lines 109-110, it looks like a filter is applied after rewriting, but it's unclear how many documents are actually useful after this process. Providing some statistics on the number of useful documents and the increase in pretraining data size would be helpful.

The role of meta-cognitive abilities, such as identifying the task or purpose of the text, in improving text quality is not clearly explained. Specifically, it is unclear how these abilities help to address issues such as grammatical errors, factual inaccuracies, or coherence problems, which are common in low-quality texts.

---

> ### Author Response · Authors · 2025-06-02
> **Response to Reviewer Mvmh**
>
> We thank the reviewer for their constructive feedback!
>
> **Statistics on the increase in useful training documents after filtering.** For filtering after rewriting, we only retain the top 10% of the highest scoring documents. The exact number of pretraining synthetic tokens that come from rewriting can be found under the “Dataset size” column in Table 1. We have provided further analysis of the increase in useful training *documents* in Section 4.1 of the paper. To summarize, about 81.7% of the new synthetic tokens used for pretraining come from recycling low-quality web texts that are *not* selected by DCLM-baseline.
>
> **The role of meta-cognitive abilities.** We appreciate that the reviewer recognizes that our rewriting prompt leverages LLM’s meta-cognitive abilities, which uses chain-of-thought reasoning to analyze the original documents in terms of facts, goals, etc. We experimented with several versions of the rewriting prompts at smaller data and model scales, and found that the prompts that leverage meta-cognitive abilities lead to better pre-training data (measured by downstream performance). Besides, we have checked the effectiveness of the prompt via manual inspection (see Appendix for document examples), as well as quantitative analysis. Specifically, through the semantic similarity analysis (Figure 3), we demonstrate that our pipeline goes beyond simple fixes (e.g. paraphrasing or stylistic changes) to improving the semantics of the original texts as a whole. For the rebuttal period, we have also run additional factuality evaluations, and found that mixing in the rewritten data improves truthfulness and knowledge capabilities of the resulting model. Refer to the Global Response for more details.
>
> **Missing citation.** This work came out in May after our COLM submission, we will add the citation in the next version of the paper.

---

> ### Comment · Reviewer_Mvmh · 2025-06-08
> **Reviewer response**
>
> Thank you for the reply. I would still like to keep the same score.

---

> > ### Author Response · Authors · 2025-06-10
> > **Author response**
> >
> > Thank you for acknowledging our rebuttal!

---

### Official Review · Reviewer_dApK · 2025-05-12

**Rating:** 7
**Confidence:** 3
**Ethics Flag:** 1

**Summary:**

This paper addresses the data bottleneck for training large language models by utilizing discarded low to medium quality document i.e. either by having a large LLM rephrase or rewrite the content. The authors compare to several baselines and vary various parameters that can affect the performance such as size of LLM (1B and 3B), total number of tokens seen, the starting document pool. Overall, the results show that Llama 3 70B can successfully rewrite discarded documents and improve the performance on two benchmarks MMLU and CORE. The topic the paper is addressing is very important however it is hard to follow as some of the design choices/results are not explained clearly.

**Questions To Authors:**

There are many typos in the paper including the name of the approach and the prompt.

In figure 7 low dimensional similarity graph, rephrase data seem farthest away -- rephrase implies changing the wording without changing the semantic meaning. Perhaps, it is more than rephrase?

It would be good to have 1B version of Figure 6 in the Appendix.

I would group the entries in Table 1 by dataset size.

**Reasons To Accept:**

This paper addressed an important problem of not having enough good quality data that can be used for LLM pretraining. The authors establish a rewriting approach with the help of a large LLM to demonstrate previously discarded low quality documents can be used re-used to improve LLM's performance on two benchmarks.

**Reasons To Reject:**

Some experimental design choices are not clearly motivated. For instance, the fastText classifier for quality. It is similar to DCLM classifier in regards to positive samples and number of training examples but negative examples are random synthetic generations. The training dataset choice is also not clear. One being synthetic and the other being social media. Considering majority of LLM pretraining data is from Wikipedia, books etc, it is not clear these would make good positive examples. DCLM classifier seems to be available on huggingface so what is the motivation on training a new one?

Another unclear point is once it is determined some documents are not paraphrased but actually rewritten, the hallucination possibility could have been analyzed and not just mentioned in limitations.

The improvements on performance are not great e.g. for 3B, the best performance is achieved after applying a mixing ratio. Once that is considered for the proposed approach, similar ratios should have been considered for other setups where it is applicable. For 3B, if that setup is removed, the best performing model for CORE becomes Raw text (top 10%), 2×. Table 1 has several of these not easily explainable results for 3B model such as PreSelect, 2× having less performance than PreSelect on MMLU or that Rewritten Text 10% performs better at MMLU without raw text being added.

---

> ### Author Response · Authors · 2025-06-03
> **Response to reviewer dApK**
>
> Thank you for the valuable suggestions!
>
> **Why train a new fastText classifier?** The DCLM fastText classifier was trained on the natural text distribution (scraped from Common Crawl) and thus its discriminative/ filtering capability is hampered when deployed on our synthetic text distribution (i.e. texts that are considered out-of-distribution). As a result, we have to retrain the fastText classifier. We follow the DCLM classifier’s training recipe by using the same datasets (OpenHermes 2.5 + ELI5) for positive examples, but replacing the negative examples to be i.i.d. samples from our synthetic data pool before any filtering. This allows us to stay as close to the DCLM setup as possible, isolating the data distribution (raw/ synthetic/ mixed) as the main factor influencing downstream performance.
>
> **Choice of training dataset for classifier.** We assume that by "one being synthetic and the other being social media", you are referring to the positive examples used to train the fastText classifier. We agree that in general, improvements in filter design can improve the quality of the synthetic data further. Our choice of positive examples simply follows the DCLM paper as we want to keep changes in the filtering pipeline to a minimum. We note that the DCLM paper did experiment with various choices of positive examples (see Table 5 of the paper), including Wikipedia, and found that using OH-2.5 + ELI5 led to the best filtered datasets. Therefore, we adopt the same design choice.
>
> **Analysis of hallucination.** It is true that there are cases where the rewriting pipeline elaborates on the existing content and adds in details (which themselves could be new facts). We find that defining a knowledge base against which hallucination is measured is not tractable in our setup. Furthermore, a lot of existing hallucination detection benchmarks use LLMs as part of the pipeline, models that are themselves susceptible to hallucination. As a proxy to measure the impact of including rewritten content on factuality, here we compare the performance of training with and without synthetic texts, on TruthfulQA and on the World Knowledge subset of DCLM Extended tasks (Jeopardy, MMLU, BigBench QA Wikidata, BigBench Misconceptions, ARC Easy, ARC Challenge, TriviaQA). **We find that adding in rewritten data doesn’t harm, but rather improves truthfulness and knowledge capabilities of the resulting model.**
>
> |                                                   | (zero-shot) TruthfulQA MC1 Acc | (zero-shot) TruthfulQA MC2 Acc | (few-shot) DCLM’s Extended World Knowledge Evals |
> |---------------------------------------------------|--------------------------------|--------------------------------|--------------------------------------------------|
> | 3B/ Raw text (top 10%)                            | 0.185                          | 0.324                          | 0.440                                            |
> | 3B/ Raw text (top 10%) + Rewritten text (top 10%) | 0.213                          | 0.355                          | 0.457                                            |
> | 7B/ Raw text (top 10%)                            | 0.234                          | 0.361                          | 0.513                                            |
> | 7B/ Raw text (top 10%) + Rewritten text (top 10%) | 0.266                          | 0.406                          | 0.543                                            |

---

> > ### Author Response · Authors · 2025-06-03
> > **Response to Reviewer dApK (continued)**
> >
> > **Performance improvement from data mixing.** The reviewer mentioned that without mixing weights, "the best performing model for CORE becomes Raw text (top 10%), 2×". While this is true, we note that one advantage of mixing in synthetic data is having more data distributions and thus more knobs to tune in terms of pre-training data composition. Here we offer additional results for similar mixing ratios applied to other synthetic data baselines. We find that using the data generated from our pipeline still leads to better performance compared to using Nemotron-CC’s synthetic data ([1]).
> >
> > | 3B-1x settings                                                       | MMLU  | CORE  |
> > |----------------------------------------------------------------------|-------|-------|
> > | Raw text (top 10%) x 0.6 + Rewritten text (top 10%) x 0.4            | 0.274 | 0.375 |
> > | Raw text (top 10%) x 0.6 + Nemotron-CC HQ extracted knowledge x 0.4  | 0.261 | 0.364 |
> > | Raw text (top 10%) x 0.6 + Nemotron-CC MQ Wikipedia rephrasing x 0.4 | 0.268 | 0.368 |
> >
> > For more evidence on the effectiveness of our approach, we have also posted additional results at a larger scale (7B model, 138B tokens seen) with no mixing ratio. There, our findings from the paper still hold: (i) adding carefully filtered rewritten texts to the high-quality web-scraped pretraining set boosts performance on average and especially on MMLU, and (ii) combining high-quality raw and rewritten documents outperforms having access to 2x more high-quality raw data.
> >
> > **PreSelect, 2× having less performance than PreSelect on MMLU** The MMLU accuracies for both of these baselines do not improve above random chance; we grayed out random-chance-level accuracies in Table 1 and comparisons should not be made about these numbers.
> >
> > **Rewritten Text 10% performs better at MMLU without raw text being added.** This is true, we find that the high-quality rewritten texts are especially helpful for improving MMLU performance, despite not being rewritten to have MMLU format or topics. We hypothesize that mixing two datasets, one being good at MMLU (synthetic texts) while the other is less so (raw texts), could dilute the performance of the former. This possibly explains the phenomenon that the reviewer brought up.
> >
> > **More than rephrasing.** We assume you are referring to Figure 3 instead of Figure 7. The figure indeed illustrates the exact point that you mentioned: our proposed pipeline does more than rephrasing, as in some cases we observe that the semantic similarity of the original document and the rewritten version is much lower than what can be expected from simple rephrasing. We provided more of such discussion in Section 4.2 of our paper.
> >
> > **Typos/ Grouping entries in Table 1.** Thanks for the suggestions, we will incorporate the corresponding changes in the next iteration of the paper.
> >
> > [1] Su, Dan, et al. "Nemotron-CC: Transforming Common Crawl into a Refined Long-Horizon Pretraining Dataset." arXiv preprint arXiv:2412.02595 (2024).

---

> > > ### Comment · Reviewer_dApK · 2025-06-10
> > >
> > > Thanks for the detailed response and clarifications on your design choices. If possible, I recommend including the motivation behind your experimental design choices in the paper as well. Considering also the substantial amount of work required to conduct the ablation studies for the pretraining task, I have increased my score.

---

> > > > ### Author Response · Authors · 2025-06-10
> > > > **Author response**
> > > >
> > > > Thank you for acknowledging our rebuttal and for adjusting the rating! Per your suggestion, we will clarify the motivation behind our experimental design further in the next version of the paper.

---

### Author Response · Authors · 2025-06-03
**Global response (with new results)**

We would like to thank all reviewers again for providing constructive feedback and for recognizing that (i) our work addresses an important and timely issue of high-quality LLM pre-training data running out (Reviewers dApK, Mvmh, NFsi), (ii) our experiments are rigorous and comprehensive (Reviewers Mvmh, NFsi, VaHx), and (iii) our analysis and ablations are detailed (Reviewers Mvmh, NFsi, VaHx) .

To address some uncertainty about the performance gains (Reviewers dApK, NFsi), here we provide additional results showing that the gain from our method increases with scale. Specifically, at the 7B-1x scale, we observe that mixing high-quality raw texts (DCLM-baseline) and our high-quality rewritten texts with 1:1 ratio leads to 12% improvement on MMLU and 2.5% improvement on average across 22 tasks of DCLM's CORE, compared to training on only the former. Similar to the observations at 1B and 3B scales, adding synthetic tokens to the pre-training mix also outperforms training on 2x more DCLM-baseline tokens (i.e. as if we had access to a bigger pool with 2x more web data). The same findings hold for the 7B-2x scale. We provide a breakdown of the per-task performance differences among the different 7B-1x baselines in https://postimg.cc/JD0GtNjj. We observe that "Raw text (top 10%) + Rewritten text (top 10%)" matches or outperforms "Raw text (top 10%)" on 19 out of 23 tasks. These results indicate that our method is an effective and scalable approach to address the "data wall" of scaling pre-training.

|                                               | Pool size | Dataset size  | MMLU  | CORE  |
|-----------------------------------------------|-----------|---------------|-------|-------|
| **7B-1x/ 138B tokens seen (2-4 epochs)**          |           |               |       |       |
| Raw text (top 10%)                            | 345B      | 34.5B         | 0.326 | 0.420 |
| Raw text (top 10%) + Rewritten text (top 10%) | 345B      | 34.5B + 34.5B | 0.447 | 0.445 |
| Raw text (top 10%), 2x                        | 690B      | 69B           | 0.356 | 0.425 |
| **7B-2x/ 276B tokens seen (4-8 epochs)**          |           |               |       |       |
| Raw text (top 10%)                            | 345B      | 34.5B         | 0.426 | 0.456 |
| Raw text (top 10%) + Rewritten text (top 10%) | 345B      | 34.5B + 34.5B | 0.499 | 0.479 |
| Raw text (top 10%), 2x                        | 690B      | 69B           | 0.472 | 0.474 |


\
Reviewers dApK and Mvmh had questions about the impact of our rewriting pipeline on the resulting data’s knowledge gaps and factuality. We ran some additional evaluations to specifically assess such properties. By comparing the performance of pre-training with and without synthetic texts on TruthfulQA and on the World Knowledge subset of DCLM’s Extended tasks (Jeopardy, MMLU, BigBench QA Wikidata, BigBench Misconceptions, ARC Easy, ARC Challenge, TriviaQA), we observe that adding in synthetic data doesn’t harm, but rather improves truthfulness and knowledge capabilities of the resulting models.

|                                                   | (zero-shot) TruthfulQA MC1 Acc | (zero-shot) TruthfulQA MC2 Acc | (few-shot) DCLM’s Extended World Knowledge |
|---------------------------------------------------|--------------------------------|--------------------------------|--------------------------------------------------|
| 3B/ Raw text (top 10%)                            | 0.185                          | 0.324                          | 0.440                                            |
| 3B/ Raw text (top 10%) + Rewritten text (top 10%) | 0.213                          | 0.355                          | 0.457                                            |
| 7B/ Raw text (top 10%)                            | 0.234                          | 0.361                          | 0.513                                            |
| 7B/ Raw text (top 10%) + Rewritten text (top 10%) | 0.266                          | 0.406                          | 0.543                                            |

---

### Decision · Program_Chairs · 2025-07-08

**Decision:**

Accept

**Comment:**

This paper looks at ways to transform data that was initially discarded in a filtering process for pre-training an LLM. The motivation is that we are running out of text data online so we need to use it more efficiently and intelligently. Overall, the reviewers were fairly positive for the paper and many of them acknowledged the timeliness of the paper as well as the novelty and importance.

Some of the reasons to accept as given by the reviewers:
•	“This paper addressed an important problem of not having enough good quality data that can be used for LLM pretraining. The authors establish a rewriting approach with the help of a large LLM to demonstrate previously discarded low quality documents can be used re-used to improve LLM's performance on two benchmarks.”
•	“The experiments are thorough and conducted under wide variety of settings, which are crucial in understanding the effectiveness of the approach. The analysis sections are also comprehensive.”
•	“The paper studies a very important and timely topic, as high-quality data is soon becoming consumed, the need for synthesizing high-quality data, as well as transforming mid-to-low quality to high quality data is needed.
•	“The paper's experiment setting is rigorous, containing detailed description and many different baselines. The results are able to outperform the strong DCLM baseline”